Comparative analysis of the metabolically active microbial communities in the rumen of dromedary camels under different feeding systems using total rRNA sequencing

Rabee Alaa Emara 1 rabee_a_m@yahoo.com
Forster Robert 2
Elekwachi Chijioke 2
http://orcid.org/0000-0002-7239-8033 Sabra Ebrahim 3
Lamara Mebarek 4
1 Animal and Poultry Nutrition Department, Desert Research Center , Cairo, Cairo , Egypt
2 Lethbridge Research and Development Centre, Agriculture and Agrifood Canada , Lethbridge, AB , Canada
3 Genetic Engineering and Biotechnology Research Institute, University of Sadat City , Sadat City, Menoufia , Egypt
4 Institut de Recherche sur les Forêts, Université du Québec en Abitibi-Témiscamingue , Rouyn-Noranda, QC , Canada
LaMontagne Michael
Electronic publication date: 2020 Oct 29
Publication date: 2020
Volume: 8
Electronic Location ID: e10184
Received 2019 Dec 17; Accepted 2020 Sep 23
Copyright: © 2020 Rabee et al.
Copyright year: 2020
Copyright holder: Rabee et al.
License: This is an open access article distributed under the terms of the Creative Commons Attribution License, which permits unrestricted use, distribution, reproduction and adaptation in any medium and for any purpose provided that it is properly attributed. For attribution, the original author(s), title, publication source (PeerJ) and either DOI or URL of the article must be cited.
License URL: https://creativecommons.org/licenses/by/4.0/

Keywords: Arabian camel, Rumen, Bacteria, Archaea, Fungi, Protozoa, Diversity, rRNA sequencing, Metatranscriptomics, Feeding regime

Funding: The authors received no funding for this work.

==============================
Breakdown of plant biomass in rumen depends on interactions between bacteria, archaea, fungi, and protozoa; however, the majority of studies of the microbiome of ruminants, including the few studies of the rumen of camels, only studied one of these microbial groups. In this study, we applied total rRNA sequencing to identify active microbial communities in 22 solid and liquid rumen samples from 11 camels. These camels were reared at three stations that use different feeding systems: clover, hay and wheat straw (G1), fresh clover (G2), and wheat straw (G3). Bacteria dominated the libraries of sequence reads generated from all rumen samples, followed by protozoa, archaea, and fungi respectively. Firmicutes, Thermoplasmatales, Diplodinium, and Neocallimastix dominated bacterial, archaeal, protozoal and fungal communities, respectively in all samples. Libraries generated from camels reared at facility G2, where they were fed fresh clover, showed the highest alpha diversity. Principal co-ordinate analysis and linear discriminate analysis showed clusters associated with facility/feed and the relative abundance of microbes varied between liquid and solid fractions. This provides preliminary evidence that bacteria dominate the microbial communities of the camel rumen and these communities differ significantly between populations of domesticated camels.

Introduction

Camels (Camelus dromedaries) can produce milk and meat in hot, arid and semi-arid regions and provide food security as the climate warms (Samsudin et al., 2011; Faye, 2013). Camels also provide textiles (fiber and hair) and are commonly used for transportation, agriculture, tourism, racing (Rabee et al., 2019). The unique feeding behavior and the functional structure of digestive tract of these pseudo-ruminants is well adapted to deserts (Kay & Maloiy, 1989). The retention time of feed particles in the camel forestomach is longer than the retention time for true ruminants, which improves the efficiency of digestion (Lechner-Doll & Engelhardt, 1989). The feed ranchers provide camels, which ranges from forage in traditional pastures to concentrated supplements in intensive feedlots, influences the structure of the camel microbiome (Faye, 2013; Henderson et al., 2015).

The chemical composition of diet shapes fermentation in rumen. For instance, cellulolytic and hemicellulytic diets favor the fibrolytic microorganisms; while, starch and sugars favor the amylolytic (Carberry et al., 2012). Also, the microbial composition and diversity varies between liquid and solid rumen fractions, which might indicate different roles in rumen fermentation; for instance, plant-adherent microbiota might have a major role in fiber degradation (Ren et al., 2020).

Digestion in the camels depends on microbial fermentation in rumen (Samsudin et al., 2011) and the efficiency of this microbial fermentations is based on the interactions between a wide variety of microbial groups, including bacteria, archaea, fungi and protozoa (Yanagita et al., 2000; Kamra, 2005). Analysis of these microbial communities could lead to increases in animal productivity and reduction of greenhouse gas emissions (Henderson et al., 2015). Unlike other ruminants, camels can utilize thorny and low quality plants like shrubs with high lignocelulolytic content (Samsudin et al., 2011). Consequently, camel rumen microbes must have the capacity to degrade such poor-quality feeds (Gharechahi et al., 2015). However, the microbial community in the rumen of dromedary camel received less attention than other domesticated ruminants.

Recent development of next generation sequencing technologies provide a rapid method of microbial identification in rumen and overcome the intrinsic constraints of traditional culture-based methods (Samsudin et al., 2011; Ishaq & Wright, 2014). Most of assessments of microbial groups in the rumen have relied on amplicon sequencing, which target a specific variable region on 16S rRNA gene (Li et al., 2016). This approach needs a wide range of primers to study different microbial communities (Kittelmann et al., 2013). Therefore, primer selection and amplification conditions could bias the output (Guo et al., 2015; Li et al., 2016; Elekwachi et al., 2017).

Total RNA sequencing (RNA-Seq) offers the advantage of specifically targeting active microbes and avoids biases associated with primer selection and chimera generation in PCR (Gaidos, Rusch & Ilardo, 2011; Guo et al., 2015; Li et al., 2016). In addition, RNA-Seq approach is capable of identifying novel microbes as it is not reliant on primers for known microbes (Li et al., 2016). High-throughput metatranscriptomic sequencing provides a comprehensive understanding of biological systems by characterization of different groups of organisms in the same environment based on the sequencing of coding and noncoding RNA (Elekwachi et al., 2017). Total RNA-Seq was applied to investigate microbial communities in many different systems including, for example, human gut (Qin et al., 2012), and cow rumen (Li et al., 2016; Elekwachi et al., 2017).

Previous microbiome studies on camel rumen have characterized one or two microbial groups using classical or molecular approaches. For example, the protozoal community in camel rumen was studied heavily by conventional microscopic methods (Ghali, Scott & Jassim, 2005; Baraka, 2012). Regarding the anaerobic fungi, a new fungal genus, Oontomyces was isolated from the rumen of Indian camel (Dagar et al., 2015), and only one study investigated whole fungal community in the gut of camel (Rabee et al., 2019). Only three molecular-based studies are available on the bacterial community (Samsudin et al., 2011; Bhatt et al., 2013; Gharechahi et al., 2015). Furthermore, only one study classified rumen archaea (Gharechahi et al., 2015).

In the present study, total rRNA sequencing was applied to (1) get insight into the composition of active microbiota in the rumen of camels; (2) describe the distribution of microbial groups among solid and liquid rumen fractions; (3) assessing the heterogeneity of these microbial populations within different populations of domestic camels.

Materials and Methods

Rumen samples

Rumen samples were collected from 11 adult dromedary camels reared at three stations that use different feeding systems. Camels in group G1 (n = 3) were housed in the Maryout Research Station, Alexendria, Egypt and were fed on Egyptian clover hay (Trifolium alexandrinum), wheat straw and concentrates feed mixture. Camels in group G2 (n = 6) were housed at the commercial farm in the Kom Hammada and fed on fresh Egyptian clover (100% high-quality forage diet) then slaughtered in the Kom Hammada slaughterhouse, Elbehera, Egypt. Camels of group G3 (n = 2) were housed at the commercial farm in Cairo area and fed on wheat straw (100% low-quality forage diet) then were slaughtered in Pasateen slaughterhouse, Cairo, Egypt. Animals were kept on these diets for at least 1 month before the sampling time. The proximate analysis of feeds is illustrated in Table S1. Details regarding camel rumen samples in this study presented in Table S2. Rumen contents were strained immediately by two layers cheesecloth to separate the liquid and solid to form 22 samples, frozen using liquid nitrogen and stored at −80 °C before further processing (Elekwachi et al., 2017). The project was approved and all samples were collected according to the Institutional Animal Care and Use Committee, Faculty of Veterinary Medicine, University of Sadat City, Egypt (Approval number: VUSC00003).

RNA isolation, quality and quantity estimation and sequencing

The frozen rumen samples were ground using liquid nitrogen. About 0.5 g of frozen fine powder was used for total RNA isolation using Trizol-Reagent protocol (Invitrogen, Carlsbad, CA, USA), followed by RNA clean up using MEGA clear Kit (Invitrogen, Carlsbad, CA, USA). Total RNA quality and quantity were estimated using an Agilent 2100 bioanalyzer (Agilent Technologies, Santa Clara, CA, USA) and RNA 6000 Nano kit (Agilent Technologiess, Santa Clara, CA, USA, USA). One hundred nanogram of total RNA was reverse-transcribed into first strand cDNA and sequenced using Illumina rRNA MiSeq preparation kit (Illumina, San Diego, CA, USA) by Illumina MiSeq platform.

Bioinformatic data analysis

The generated RNA sequence reads were analyzed using pipeline developed by Elekwachi et al. (2017). Briefly, the sequence quality was checked using the FastQC program v. 0.11.4 (Andrews, 2010), then Trimmomatic program v. 0.35 (Bolger, Lohse & Usadel, 2014) was used to trim adaptors, barcodes, ambiguous and low quality reads. PEAR program v. 0.9.6 (Zhang et al., 2014) was used to merge read 1 and read 2 using default options. Then after, the hidden Markov models rRNA-HMM tool of the rapid analysis of multiple metagenomes with a clustering and annotation pipeline (RAMMCAP) (Li, 2009) was used to sort the reads into archaea and bacteria (16S, 23S), and eukaryote (18S, 23S) rRNA sequences. Merged sequence files were then sub-sampled as needed using MEME program v. 4.10.2 (Bailey et al., 2009). For each sample, 70,000 reads were run through the pipeline. For subsequent analysis steps, 20,000, 10,000, and 2,000 sequences were used for bacteria, eukaryote and archaea, respectively. Taxonomy binning for eukaryote and archaeal SSU rRNA sequences was performed using BLASTN. The sub-sampled query sequences were searched against the SILVA SSURef-111 database using an e-value of 1e−5. Bacterial SSU sequences were binned into operational taxonomic units (OTUs) using the “classify. seqs” command of Mothur v. 1.33.1 program (Schloss et al., 2009). The SSURef-108 gene and the SSURef-108b taxonomy databases were used. Principal co-ordinate analysis (PCoA) using Bray Curtis dissimilarity and alpha diversity indices (Chao1, Shannon and Inverse Simpson) were evaluated by Mothur (Schloss et al., 2009) based on sub-sampling of 70,000 reads per sample according the protocol “Community Structure Analysis Based on OTU Clustering” outlined in Elekwachi et al. (2017).

Statistical analyses

Data of relative abundance of bacterial phyla, protozoal genera, fungal genera and archaea genera and order Thermoplasmatales were tested for normality and homogeneity using Shapiro–Wilk test and variables that were deemed non-normal were then arcsine transformed. Linear Discriminate Analysis (LDA) and Bray Curtis Permutational Multivariate Analysis of Variance (PERMANOVA) tests depended on the relative abundance of bacterial phyla. All the protozoal, fungal and archaeal genera and the order Thermoplasmatales were used to show the differences in community structure and to compare the clustering of samples. Pearson correlation analysis was used to identify correlation within and between microbial communities and the correlation scores were visualized as a heatmap. The statistical analyses were performed using the SPSS v. 20.0 software package (SPSS, 1999) and PAST (Hammer, Harper & Ryan, 2001). Sequences were deposited to the sequence read archive (SRA) under the accession number: SRP107370.

Results

The composition and diversity of active microbial community

Total rRNA sequencing in 22 solid and liquid rumen samples from 11 camels resulted in a total of 3,958,591 reads with average of 359,872 ± 85,366 (mean ± standard error (SE)) reads per animal in the solid fraction (SF) and 3,386,392 reads with an average of 307,854 ± 60,989 reads per animal in the liquid fraction (LF). The sequence reads of bacteria dominated the active microbial community, followed by protozoa, archaea and fungi (Table 1). Relative abundance of protozoa was higher in liquid fraction of G1 (LF-G1), while relative abundance of bacteria was higher in solid fraction of G1 (SF-G1). The highest population of archaea was observed in G2 camels. Additionally, G3 camels showed the highest relative abundance of fungi (Table 1; Fig. S1). Number of OTUs and Alpha-diversity indices, Chao1, Shannon and Inverse Simpson, were higher in the rumen of LF-G2 samples (Table 1).

Table 1 The relative abundance (%) of bacteria, archaea, protozoa and fungi and diversity indices.

The relative abundance (%) of bacteria, archaea, protozoa and fungi and OTU numbers and values of Shannon, Chao1 and Inverse Simpson indices in the ruminal solid (SF) and liquid (LF) fractions of dromedary camels fed a mixed ration (G1), high-quality forage (G2) and low-quality-forage (G3) (Mean ± Standard error (SE)).

Item	G1	G2	G3	Overall mean	
Bacteria SF	92 ± 1	89 ± 2	89 ± 2	90 ± 1	
Bacteria LF	85 ± 4	91 ± 2	87 ± 8	88 ± 2	
Archaea SF	2.3 ± 0.2	3.4 ± 0.4	2.2 ± 1.0	3.0 ± 0.3	
Archaea LF	2.2 ± 0.2	2.8 ± 0.4	1.8 ± 0.2	2 ± 0.3	
Protozoa SF	5 ± 1	7 ± 2	6 ± 2	6 ± 1	
Protozoa LF	12 ± 4	6 ± 1.6	8 ± 5	8 ± 1.6	
Fungi SF	0.15 ± 0.05	1 ± 0.3	3 ± 1	1 ± 0.4	
Fungi LF	0.35 ± 0.1	0.5 ± 0.1	3 ± 3	1 ± 0.5	
OTUs SF	1,012 ± 43	1,201 ± 38	1,135 ± 148	1,137 ± 39	
OTUs LF	1,076 ± 26	1,229 ± 38	1,147 ± 53	1,172 ± 30	
Shannon SF	6 ± 0.1	7 ± 0.10	7 ± 0.3	7 ± 0.1	
Shannon LF	6.5 ± 0.06	7 ± 0.1	7 ± 0.1	7 ± 0.1	
Chao1 SF	6,644 ± 650	9,329 ± 714	9,028 ± 1,985	8,542 ± 608	
Chao1 LF	7,280 ± 521	10,839 ± 724	7,688 ± 625	9,295 ± 672	
Invsimpsone SF	117 ± 14	863 ± 306	644 ± 398	620 ± 196	
Invsimpsone LF	13 5± 21	983 ± 492	612 ± 142	684 ± 282	

Bacterial community

The composition of bacterial community varied little between groups and consisted of 12 phyla. The five most predominant phyla were Firmicutes, Bacteroidetes, Proteobacteria, Spirochaetes and Fibrobacteres, respectively (Table 2). Firmicutes dominated the bacterial community in all groups and was higher in G2 followed by G1 and G3 camels, respectively, and was also higher in SF compared to LF (Table 2). At the family level, Lachnospiraceae and Ruminococcuceae dominated the Firmicutes. In addition, six genera dominated this phylum, including Butyrivibrio, RFN8-YE57, Ruminococcus, vadinHA42, Acetitomaculum and Blautia (Fig. 1A; Table S3). The second largest phylum, Bacteroidetes, showed the highest relative abundance in G3 followed by G1 and G2 camels and was higher in LF than SF (Fig. 1A; Table S3). At the family level, Prevotellaceae, BS11_ gut_ group, and Rikenellaceae dominated the Bacteroidetes; and at the genus level, Prevotella, RC9_gut_group dominated the Bacteroidetes. Proteobacteria, phylum showed a higher relative abundance in LF-G1 samples and was dominated by Succinivibrionaceae family and Desulfovibrio genus (Table 2; Fig. 1A; Table S3). The Spirochaetes phylum was higher in the SF-G3 and it was classified into two families including Spirochaetaceae and PL-11B10 and was dominated by Treponema genus. Fibrobacteres phylum was higher in SF-G3 (Table 2; Fig. 1A; Table S3). Actinobacteria were higher in SF-G2 samples, Tenricutes phylum was higher in LF-G1 samples and Lentisphaerae phylum, was about 3-fold higher in LF as relative to SF and accounted for a large population in the camels of G3 (Table 2). Additionally, several minor bacterial phyla were also observed in the rumen of camels such as Verrucomicrobia, Elusimicrobia, Cyanobacteria and Chloroflexi (Table 2).

Table 2 Relative abundance (%) of bacterial phyla.

Relative abundance (%) of bacterial phyla in the ruminal solid (SF) and liquid (LF) fractions of camels fed a mixed ration (G1), high-quality forage (G2) and low-quality forage (G3) (Mean ± Standard Error (SE)).

Bacterial Phylum	G1	G2	G3	Overall mean	
Firmicutes SF	63 ± 2	65 ± 0.1	48 ± 10	60 ± 3	
Firmicutes LF	46 ± 3	56 ± 2	45 ± 13	50 ± 3	
Bacteroidetes SF	20 ± 1	15 ± 1	27 ± 8	19 ± 2	
Bacteroidetes LF	31 ± 0.5	21 ± 1.5	31 ± 12	26 ± 3	
Proteobacteria SF	5 ± 1	3.5 ± 0.3	3 ± 0.5	4 ± 0.3	
Proteobacteria LF	6.5 ± 1	6 ± 2	3 ± 0.1	5.5 ± 1	
Spirochaetes SF	3 ± 0.6	5 ± 1	6 ± 1.5	4.5 ± 0.6	
Spirochaetes LF	3.7 ± 1	2.6 ± 0.5	5.6 ± 1	3.5 ± 0.5	
Fibrobacteres SF	2.5 ± 0.6	4 ± 0.7	9 ± 1	4.5 ± 1	
Fibrobacteres LF	1.6 ± 0.5	2.5 ± 1	7 ± 3	3 ± 1	
Actinobacteria SF	2 ± 0.2	4.5 ± 0.3	1.5 ± 0.3	3 ± 0.5	
Actinobacteria LF	1.5 ± 0.14	5.5 ± 1	1 ± 0.1	3.6 ± 10	
Lentisphaerae SF	0.7 ± 0.03	0.7 ± 0.1	1.5 ± 0.2	1 ± 0.1	
Lentisphaerae LF	3.2 ± 0.3	2 ± 0.5	3.2 ± 2	2.6 ± 0.4	
Tenericutes SF	2 ± 0.4	1 ± 0.1	0.6 ± 0.3	1 ± 0.2	
Tenericutes LF	3.7 ± 0.6	1.5 ± 0.3	0.4 ± 0.1	1.8 ± 0.4	
Verrucomicrobia SF	0.3 ± 0.1	0.20 ± 0.1	0.6 ± 0.4	0.30 ± 0.1	
Verrucomicrobia LF	2.2 ± 0.4	1 ± 0.3	1.3 ± 0.3	1.3 ± 0.3	
Chloroflexi SF	0.4 ± 0.03	0.5 ± 0.06	0.24a	0.4 ± 0.04	
Chloroflexi LF	0.3 ± 0.03	0.3 ± 0.05	0.24a	0.3 ± 0.02	
Cyanobacteria SF	0.3 ± 0.04	0.3 ± 0.05	0.5a	0.35 ± 0.04	
Cyanobacteria LF	0.3 ± 0.05	0.3 ± 0.05	0.25a	0.3 ± 0.03	
Elusimicrobia SF	0.2 ± 0.05	0.15	0.3 ± 0.14	0.2 ± 0.04	
Elusimicrobia LF	0.3 ± 0.07	0.2 ± 0.04	0.8 ± 0.4	0.4 ± 0.1	
Note:

a The value was calculated from one animal.

Figure 1 The relative abundance of microbial groups.

Comparison of relative abundance of genera of the microbiota in dromedary camel. bacterial (A), archaeal (B), protozoal (C) and fungi (D) in ruminal solid (SF) and liquid (LF) fractions of camels under different feeding systems.

Of the 74 genera observed, only seven were observed exclusively in libraries generated from a specific facility, including uncultured Marinilabiaceae (Bacteroidetes), Quinella (Firmicutes) and Streptococcus (Firmicutes) that were observed only in G2 and G3 camels. Ruminobacter (Proteobacteria) was observed only in G1 and G2 camels. On the other hand, Arcobacter and Succinivibrio within phylum Proteobacteria were observed only in G1 camels and Betaproteobacteria (Proteobacteria) was observed only in G3 camels. Moreover, many unclassified bacteria were observed across samples and accounted for 39% of total bacterial reads. Most of these unclassified bacterial reads were observed in phylum Firmicutes and Bacteroidetes.

Archaeal community

Reads that classified as archaea were further classified to three orders within the phylum Euryacheota: Thermoplasmatales, Methanobacteriale and Methanomicrobial. Thermoplasmatales dominated the archaeal community and showed the highest population in LF-G3 samples, this order was not classified out of order level (Table 3; Fig. 1B). Reads that classified in the Methanobacteriale were further classified to family Methanobacteriacea that includes three genera: Methanobrevibacter, Methanophera and Methanobacterium. Methanobrevibacter is the second largest contributor in archaeal population and was higher in SF-G1 samples. Methanosphaera exhibited higher relative abundance in SF-G2 samples. Methanobacterium was absent in G3 camels; however, a small proportion of this genus was found in the camels of G1 and G2. Methanomicrobium genus, which belongs to order Methanomicrobiales and family Methanomicrobiaceae was the least contributor in archaeal population and was more prevalent in LF-G3 samples (Table 3; Fig. 1B).

Table 3 Relative abundance (%) of archaeal orders and genera.

Relative abundance (%) of archaeal orders and genera observed in the ruminal solid (SF), and liquid (LF) fractions of camels under different feeding systems. Animals in G1 fed a mixed ration, animal in G2 fed high-quality forage and animal in G3 fed low quality-forage (Mean ± Standard Error (SE)).

Archaea	G1	G2	G3	Overall mean	
Thermoplasmatales SF	33 ± 7	33 ± 4	55 ± 10	37 ± 4	
Thermoplasmatales LF	46 ± 8	48 ± 3	67 ± 5	51 ± 3	
Methanomicrobium SF	1 ± 0.3	0.3 ± 0.2	8 ± 1	2 ± 0.9	
Methanomicrobium LF	2 ± 0.5	1 ± 0.5	9 ± 6	3 ± 1	
Methanobrevibacter SF	51 ± 5	42 ± 3	34 ± 9	43 ± 3	
Methanobrevibacter LF	43 ± 5	39 ± 2.4	23 ± 0.01	37 ± 2	
Methanosphaera SF	15 ± 2	24 ± 3	3 ± 1	18 ± 3	
Methanosphaera LF	8 ± 2	12 ± 1.5	2.5 ± 1	9.5 ± 1.5	
Methanobacterium SF	0.05	0.06	0	ND	
Methanobacterium LF	0.2 ± 0.02	0.1 ± 0.02	0	ND	
Note:

ND: Non Determined.

Protozoal community

Reads that classified as protozoa were further classified to two families: Ophryoscolecidae and Isotrichidae (Table 4). Reads that classified in the Ophryoscolecidae were further classified to seven genera, Diplodinium, Ophryoscolex, Entodinium, Polyplastron, Eudiplodinium, Epidinium and Trichostomatia. Reads that classified in the Isotrichidae were further classified to two genera, Dasytricha and Isotricha. The variation among the camels in protozoal population was clearly observed and seemed to be higher than other microbial communities; however, the protozoal community composition was similar among the camels (Table 4; Fig. 1C). The most dominant protozoal genera were Diplodinium, Ophryoscolex and Entodinium. Camels in G1 had the highest population of Entodinium and Epidinium. Camels in G2 had the greatest population of Eudiplodinium, Ophryoscolex, Isotricha and Dasytricha and camels in G3 had the greatest population of Diplodinium, Polyplastron and Trichostomatia. On the sample fraction level, solid fraction had a higher representation of Ophryoscolex, Polyplastron, Eudiplodinium, Epidinium and Diplodinium, while liquid fraction had a higher representation of Entodinium, Isotricha and Dasytricha (Table 4; Fig. 1C).

Table 4 Relative abundance (%) of protozoal genera.

Relative abundance (%) of protozoal genera in the ruminal solid (SF) and liquid fraction (LF) of camels under different feeding systems. Animals in G1 fed a mixed ration, animals in G2 fed high-quality forage and animals in G3 fed low-quality forage (Mean ± SE).

Protozoa	G1	G2	G3	Overall mean	
Entodinium SF	23 ± 6	6.5 ± 0.6	6 ± 1	11 ± 3	
Entodinium LF	54 ± 10	15 ± 2.5	5 ± 1	24 ± 6	
Polyplastron S F	10 ± 1	17.5 ± 2	25 ± 3	17 ± 2	
Polyplastron LF	6 ± 1	11 ± 0.2	24 ± 3	12 ± 2	
Diplodinium S F	23 ± 1	35 ± 3	49 ± 10	34 ± 3	
Diplodinium LF	13 ± 3	27 ± 3	61 ± 6	29 ± 5	
Eudiplodinium SF	8 ± 0.6	8 ± 2	2 ± 0.7	7 ± 1	
Eudiplodinium LF	4 ± 1	5.5 ± 1	2.5 ± 0.5	4.5 ± 0.6	
Epidinium SF	5 ± 0.8	4 ± 1	2 ± 1	4 ± 0.1	
Epidinium LF	3 ± 0.8	4.5 ± 0.6	1 ± 0.7	3.5 ± 0.5	
Ophryoscolex SF	30 ± 4	27 ± 3	15 ± 5	26 ± 2.5	
Ophryoscolex LF	19 ± 4	29 ± 0.6	6.5 ± 4	22 ± 3	
Trichostomatia SF	0.1 ± 0.02	1 ± 0.25	0.3 ± 0.15	1 ± 0.2	
Trichostomatia LF	0.2 ± 0.04	1 ± 0.2	1 ± 0.1	1 ± 0.2	
Isotricha SF	0.2 ± 0.04	0.3 ± 0.05	0.3 ± 0.004	0.3 ± 0.03	
Isotricha LF	0.5 ± 0.2	2 ± 0.9	0.3 ± 0.01	1 ± 0.5	
Dasytricha SF	0.04 ± 0.01	1.5 ± 0.3	0.2 ± 0.15	1 ± 0.3	
Dasytricha LF	0.1 ± 0.002	5.5 ± 0.8	0.5 ± 0.3	3 ± 1	

Anaerobic rumen fungal community

Reads that classified as rumen fungi were further classified to two phyla: Neocallimastigomycota and Chytridiomycota. Reads that classified in the Neocallimastigomycota were further classified to family Neocallimasticeceae that includes three genera, Neocallimastix, Piromyces and Cyllamyces. Neocallimastix dominated the fungal community, followed by Piromyces and Cyllamyces (Table 5; Fig. 1D). These anaerobic fungal genera represented >99.5% of the fungal population. In addition, reads that classified in the Chytridiomycota were further classified to family Spizellomycetaceae that includes genus Spizellomyces, which was noted in a very small proportion (<0.5%) (Table 5). Neocallimastix was more abundant in the SF-G1 samples while Piromyces and Cyllamyces were more abundant in LF-G2 and SF-G3 respectively (Table 5; Fig. 1D).

Table 5 Relative abundance (%) of fungal genera.

Relative abundance (%) of fungal genera in the ruminal solid (SF) and liquid fraction (LF) of camels under different feeding systems. Camels in G1 fed a mixed ration, animals in G2 fed high-quality forage, and animals in G3 fed low-quality forage (Mean ± SE).

Fungi	G1	G2	G3	Overall mean	
Spizellomyces SF	0	0.1	0.02	ND	
Spizellomyces LF	0.3 ± 0.1	0.3 ± 0.1	0	ND	
Cyllamyces SF	2 ± 0.6	3 ± 1.5	7 ± 4	3.5 ± 1	
Cyllamyces LF	2 ± 0.8	3 ± 0.8	10 ± 1	4 ± 1	
Piromyces SF	6 ± 3	12 ± 0.7	8 ± 1	9 ± 1	
Piromyces LF	6 ± 4	12 ± 2	10 ± 6	10 ± 2	
Neocallimastix SF	92 ± 3	85 ± 1	85 ± 3	87 ± 1	
Neocallimastix LF	92 ± 4	85 ± 1.5	81 ± 7	86 ± 2	
Note:

ND: Non Determined.

Effect of feeding system and facility on the composition of microbial communities

Multivariate analysis separated libraries by feeding system and housing facility distinctly (Figs. 2 and 3). Also, bacteria, dominated by Firmicutes, drove differences between animals (Fig. 3). Furthermore, Entodinium, Thermoplasmatales, Neocallimastix drove differences in protozoal, archaeal and fungal communities, respectively. PERMANOVA analysis revealed that the difference between camel groups was significant (P < 0.01) in all microbial groups (Table S4). Pairwise comparison between camel groups based on Bonferroni-corrected P-value demonstrated that the difference was significant (P < 0.05) between camels of G2 and G3 in bacterial and archaeal communities (Table S4). Moreover, the difference was significant between the three groups in the protozoal community (P < 0.05), whereas, in the fungal community, the difference was significant only between camels in group G1 and G2 (Table S4).

Figure 2 Principal Co-ordinated analysis.

Principal Co-ordinated analysis derived from OTUs from 22 ruminal liquid (LF) and solid (SF) samples distributed on three camel groups. G1 camels (red circles), G2 (white circle) and G3 (blue circles).

Figure 3 Linear Discriminant analysis.

Linear Discriminant analysis of microbial communities in the samples based on the relative abundance of genera of active bacteria (A), archaea (B), protozoa (C) and fungi (D) in ruminal solid (SF), and liquid (LF) fractions of camels under three feeding systems, G1 (black dots), G2 (blue squares) and G3 (coral triangles).

Pearson correlation between microbes in the rumen of dromedary camel

Pearson correlation analysis (Figs. 4A and 4B), revealed many significant positive and negative correlations (P < 0.05). For example, in active bacteria, Bacteroidetes correlated positively with Cyllamyces and negatively with Butyrivibrio, Methanosphaera and Trichostomatia. Prevotellaceae correlated positively with Neocallimastix and Entodinium and negatively with Ruminococcaceae, Methanosphaera and Diplodinium. Fibrobacteres correlated positively with Cyllamyces, Methanomicrobium, Thermoplasmatales and Diplodinium and negatively with Methanosphaera, Epidinium, Ruminococcaceae and Butyrivibrio. Firmicutes correlated positively with Methanosphaera and negatively with Piromyces, Thermoplasmatales and Methanomicrobium.

Figure 4 Heatmap based on Pearson correlation.

Heatmap based on Pearson correlation coefficients between and within the relative abundance of bacteria, archaea, protozoa and fungi in solid (A) and liquid (B) rumen fractions of dromedary camel. The black boxed ellipses refer to the significant correlations at P < 0.05.

In active archaea, Thermoplasmatales correlated positively with Diplodinium and negatively with Methanobrevibacter and Methanosphaera. In active protozoa, there was a negative correlation between Polyplastron, Entodinium, Ophryoscolex and Epidinium. In active fungi, a negative correlation was observed between Cyllamyces, Neocallimastix and Piromyces and between Piromyces and Entodinium.

Discussion

Rumen microbes can ferment a wide variety of feed components, including cellulose, xylan, amylose and protein and produce volatile fatty acids that provide the animal with approximately 70% of daily energy requirements (Bergman, 1990; Henderson et al., 2015). Fermentation by rumen microbes also generates methane, which contributes to global warming and represents 2–12% loss of feed energy for the animal (Johnson & Ward, 1996; Carberry et al., 2012; Jami, White & Mizrahi, 2014). Investigation of these microbial communities could improve our understanding of their function in fiber digestion and lead to practices that maximize the efficiency of ruminal fermentation and minimize greenhouse gas release (Lee et al., 2012).

In this study, camel groups were fed different diets and reared in different locations. The diversity and relative abundance of microbial communities varied between camel groups, which was supported by the results of PCoA, LDA and PERMANOVA analyses. This result agrees with the results of studies of other ruminants (Henderson et al., 2015). Camels in the present study were fed on different forages; Egyptian clover and wheat straw (Table S1). Egyptian clover is the most balanced and nutritious fodder widely used for feeding camels (Carberry et al., 2012; Bakheit, 2013; Shrivastava et al., 2014), which might supported the high microbial diversity in G2 camels compared to other groups (Table 1). This was consistent with previous studies on cows (Pitta et al., 2010; Shanks et al., 2011; Kumar et al., 2015). Highly degradable carbohydrates support bacterial and protozoal growth (Dijkstra & Tamminga, 1995; Kumar et al., 2015), which could demonstrate their higher population in G1 camels. Additionally, higher bacterial population slows the fungi growth (Stewart, Duncan & Richardson, 1992; Orpin & Joblin, 1997), which was illustrated by low fungal population in G1 camels.

Bacterial community

Firmicutes phylum was more abundant than Bacteroidetes and both phyla comprised >75% of all bacterial reads (Table 2), which agrees with studies of camels (Samsudin et al., 2011), Surti Buffalo (Pandya et al., 2010) and muskoxen (Salgado-Flores et al., 2016). The majority of Firmicutes’ members have a potential role in fiber digestion, which might illustrate their higher population in G2 camels that were fed on high-quality forage and also in solid fraction. The high proportion of Ruminococcaceae and Lachnospiraceae supports this speculation (Pitta et al., 2014a; Nathani et al., 2015). Blautia and Acetitomaculum genera have a key role as reductive acetogens (Le Van et al., 1998; Yang et al., 2016) and varied among the camel groups in this study. This supports the observation that manipulation of diet can enhance reductive acetogenesis in rumen and minimize methanogenesis (Le Van et al., 1998).

Bacteroidetes were higher in samples collected from animals reared in the station that used low-quality feed (G3), which was similar to results on cattle (Pitta et al., 2014b). The phylum was dominated by family Prevotellaceae, which confirms Gharechahi et al. (2015). Members of Bacteroidetes possess diverse enzymes that can target cellulose, pectin and soluble polysaccharides released in the liquid phase (Mackenzie et al., 2015). Additionally, Prevotella genus produces propionate that is used for energy by the host (Nathani et al., 2015). We speculate that Bacteroidetes species contribute to the adaptation of camels to arid conditions.

The RC9_gut_group found in this study belongs to uncultured genera and was found also in the Rhinoceros hindgut (Bian et al., 2013). Unclassified Bacteroidetes specialize in lignocellulose degradation (Mackenzie et al., 2015), which could support their high proportion in G3 camels. Fibrobacteres was higher (3.1%) in this study compared to the other findings on camels (Gharechahi et al., 2015); this phylum is the principal cellulolytic bacteria in the rumen (Ransom-Jones et al., 2012; Nathani et al., 2015), which might illustrate its higher relative abundance in solid fraction and in the rumen of G3 camels that fed on wheat straw (Table 2). The members of Proteobacteria were lower in G2 and G3 camels that were fed on diet rich in fiber contents. These findings highlighted this phylum’s function as a protein-degrading bacteria (Liu et al., 2017). The abundance of Treponema was higher in the solid fraction and in G3 camels (Fig. 1A). Treponema is the dominant genus in Spirochaetes phylum and it is fiber-associated bacteria, which could indicate to its cellulytic and xylanolytic activities (Ishaq & Wright, 2012).

The dominant bacterial genera in this study were Butyriovibrio, RFN8-YE57, Ruminococcus, Prevotella, Fibrobacter, Treponema and VadinHA. These genera were higher in the SF except RFN8-YE57 compared to the LF; this finding was consistent with a study on camels (Gharechahi et al., 2015), and confirms that solid-attached microbes could play a major role in ruminal fiber digestion (Jewell et al., 2015; Noel et al., 2017).

Most of Elusimicrobia in this study

Most of Elusimicrobia observed in this study have yet to be cultured; some members of this phylum were isolated from the termite’s gut that degrades cellulose (Herlemann et al., 2009). Therefore, we speculate that this phylum has a role in fiber digestion and that might illustrate their high proportion in G3 camels. Actinobacteria observed also in the rumen of moose and some members of this phylum have acetogenic activities (Ishaq et al., 2015). Some members of Victivallis within Lentisphaerae phylum were involved in cellobiose degradation (Zoetendal et al., 2003).

Unclassified bacteria in our study (39% of total bacterial reads) were less than the percentage found in a study of muskoxen (54%) (Salgado-Flores et al., 2016). The presence of unclassified bacteria in the gut was commonly observed (Gruninger, McAllister & Forster, 2016) and could be a result of the presence of new bacteria that ferment plant biomass (Salgado-Flores et al., 2016) or related to short reads were generated from RNA-Seq (Li et al., 2016).

Archaeal community

Since some archaea produce CH4 from H2 and CO2, this phyla may control methane emission from ruminants (Hook, Wright & McBride, 2010). Additionally, acetate produced in fiber breakdown provides a methyl group for methanogenesis; therefore, alteration of diet shifts the structure of methanogen populations (Hook, Wright & McBride, 2010; Tapio et al., 2017), which could demonstrate the variation in the relative abundance of archaea between camel groups. Camels of the second group (G2) that were fed fresh clover, showed the highest archaeal population (Table 2) and archaeal community was dominated by Thermoplasmatales, a methylotrophic methanogens order (Table 3), which was consistent with the results on cattle (Carberry et al., 2014) and camels (Gharechahi et al., 2015). Thermoplasmatales produce methane from methyl amine and supplementing of animal’s diet with rapeseed oil decreases the abundance of this order, making it a high potential target in future strategies to mitigate methane emissions (Poulsen et al., 2013). The Methanobrevibacter, Methanosphaera, Methanomicrobium and Methanobacterium (Table 4) dominated the reads classified as archaea in this study, which agrees with trends reported for beef cattle (Carberry et al., 2014). Methanobrevibacter dominated the methanogens in other ruminant (Henderson et al., 2015) and was associated with high methane emissions (Tapio et al., 2017). Moreover, Methanomicrobium was higher in the camels of G3 that were fed on poor quality forage, which was similar to results of buffalo (Franzolin & Wright, 2016), and in vitro (Wang et al., 2018). In rumen, Methanomicrobium converts H2 and/or formate into CH4 (Leahy et al., 2013). The abundance of Thermoplasmatales was also negatively correlated with Methanobrevibacter, which is consistent with previous results (Danielsson et al., 2017; McGovern et al., 2017).

Protozoal community

The majority of protozoal reads were classified as Diplodinium, Ophryoscolex, Entodinium, Polyplastron, Eudiplodinium and Epidinium (Table 4). Similar findings were observed on different ruminants (Baraka, 2012). Feed appeared to influence the relative abundance of protozoa, as reported previously for cattle (Hristov et al., 2001; Weimer, 2015); however, we cannot differentiate the effects of feed from facility. Diplodinium dominated protozoal community and was prevalent in G3 camels, which highlighted the cellulolytic activity of this genus (Coleman et al., 1976). Some species of genus Diplodinium, such as Diplodinium cameli, were discovered in, and are unique to, the rumen of Egyptian camel (Kubesy & Dehority, 2002). In addition, Entodinium was higher in G1 camels that were fed on concentrates feed mixture that increase the protozoa. Also, this genus predominates rumen of camels (Selim et al., 1999; Ghali, Scott & Jassim, 2005) and cattle (Carberry et al., 2012). Moreover, Kittelmann & Janssen (2011) showed that the Polyplastron was the dominant genus in cattle. On the function level, all the genus Eudiplidinum, Epidinum and Diplodinum have cellulolytic activity (Coleman et al., 1976), whereas, Polyplastrone and Epidinium have a xylanolytic activity (Devillard et al., 1999; Béra-Maillet et al., 2005).

Anaerobic rumen fungal community

The highest fungal population was observed in the solid fraction and rumen of G3 camels (Table 1). These findings were in agreement with the results of studies stated that the fiber-based diets stimulated the fungal growth (Orpin, 1977; Roger et al., 1993; Kamra, 2005; Haitjema et al., 2014). This could explain the low fungal population in G1 camels in our study. Moreover, the longer retention time and neutral pH in camel’s forestomach (Russell & Wilson, 1996) make it more suitable for the survival of rumen fungi. Neocallimastix dominated the fungal community and was higher in the G1 camels, which was similar to other results on sheep and camels (Kittelmann et al., 2013, Rabee et al., 2019). This genus produces enzymes capable of hydrolyzing cellulose, xylan and starch (Pearce & Bauchop, 1985). Cyllamyces that was observed in small population, has the ability to degrade poor-quality feeds (Sridhar, Kumar & Anadan, 2014), which might explain its high population in solid fraction and G3 camels. Piromyces was the second dominant genus in the camel rumen of this study and produces cellulolytic and xylanolytic enzymes (Teunissen et al., 1992). Therefore, this genus was most abundant in rumen collected from the G2 group of camels. The genus Spizellomyces is closely related to Chytridiomctes (Bowman et al., 1992), and common in grassland and crop soil (Lozupone & Klein, 2002, Kittelmann et al., 2012). Thus, contamination of forages by soil could explain the presence of this fungus in camel rumen.

Correlation between rumen microbes

Interactions between rumen microbes drive feed degradation and methane formation in the rumen, which influence the animal production and the environment (Williams et al., 1994; Lee et al., 2012; Henderson et al., 2015). Positive and negative correlations were observed within and between microbial communities in this study (Fig. 4). Methanogens colonize protozoa and this relationship enhances methane formation (Newbold, Lassalas & Jouany, 1995). Additionally, fibrolytic bacteria produce hydrogen and methyl groups that methanogens use for growth (Johnson & Johnson, 1995), which demonstrated positive correlations found between Fibrobacteres and some methanogens. Also, positive correlation between methylotrophic Methanosphaera and Lachnospiraceae that has been implicated in pectin degradation and provides methanol as a substrate for the methylotrophs (Dehority, 1969). On the other hand, Prevotella is a hydrogen utilizer and produces propionate that impact the methanogenesis in the rumen negatively (Pitta et al., 2014a; Liu et al., 2017), which illustrates negative correlation between Prevotellaceae and archaea.

Since the rumen anaerobic fungi produce abundant H2 through the fermentation of carbohydrate; they can interact positively with H2 utilizers such as archaea, Prevotellaceae, Blautia and Acetitomaculum (Orpin & Joblin, 1997; Le Van et al., 1998; Yang et al., 2016; Liu et al., 2017). Additionally, anaerobic fungi penetrate plant tissue, providing an increased surface area for bacterial colonization (Orpin & Joblin, 1997), which could explain positive correlation between fungi and both Butyrivibrio and Fibrobacteres. However, some bacteria and protozoa prey on fungal zoospores (Morgavi et al., 1994), which demonstrated the negative correlation between both Neocallimastix and Piromyces with Diplodinium and Entodinium. Furthermore, Ruminococcus produces compounds that inhibit the growth of rumen fungi (Stewart, Duncan & Richardson, 1992), which supports the negative correlation between Neocallimastix and Ruminococcaceae. Polyplastron predates upon other protozoa like Epidinium, Eudiplodinium, Diplodinium, and Ostracodinium (Eadie, 1967).

Conclusions

The microbial community in camel rumen was diverse and similar in composition between the groups of camels. The majority of camel rumen microbes (bacteria, fungi, and protozoa) were fibrolytic or have a possible role in fiber digestion, which might illustrate the ability of camel to live in desert harsh conditions under poor feeds. Moreover, the structure of microbial community in rumen of camels was similar to other ruminants.

Supplemental Information

Supplemental Information 1 The chemical composition (%) of diets fed to camels under investigation.

Click here for additional data file.

Supplemental Information 2 Metadata information of rumen solid and liquid samples of camel under different feeding systems.

Click here for additional data file.

Supplemental Information 3 The relative abundance (%) of active bacterial genera in ruminal solid (SF) and liquid (LF) fractions of camels fed on mixed ration (G1), high quality forage (G2), low quality forage (G3).

Click here for additional data file.

Supplemental Information 4 Pairwise comparison of the abundance of active bacterial, archaeal, Protozoal, and fungal genera in the rumen of camels distributed on three groups based on the feeding system.

Click here for additional data file.

Supplemental Information 5 Comparison of overall relative abundance of bacteria, archaea, protozoa, and fungi in the rumen solid (SF) and liquid (LF) fractions in four camel groups (G1, G2, G3).

Click here for additional data file.

Additional Information and Declarations

Competing Interests

Author Contributions

Animal Ethics

Data Availability

The authors declare that they have no competing interests.

Alaa Emara Rabee conceived and designed the experiments, performed the experiments, analyzed the data, prepared figures and/or tables, authored or reviewed drafts of the paper, and approved the final draft.

Robert Forster conceived and designed the experiments, analyzed the data, authored or reviewed drafts of the paper, and approved the final draft.

Chijioke Elekwachi performed the experiments, analyzed the data, authored or reviewed drafts of the paper, and approved the final draft.

Ebrahim Sabra conceived and designed the experiments, performed the experiments, authored or reviewed drafts of the paper, sample collection, and approved the final draft.

Mebarek Lamara analyzed the data, prepared figures and/or tables, authored or reviewed drafts of the paper, and approved the final draft.

The following information was supplied relating to ethical approvals (i.e., approving body and any reference numbers):

The Institutional Animal Care and Use Committee, Faculty of Veterinary Medicine, University of Sadat City, Egypt approve the study (VUSC00003).

The following information was supplied regarding data availability:

Data is available at the SRA database: SRP107370.

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
