# Peer review of "Comparative analysis of the metabolically active microbial communities in the rumen of dromedary camels under different feeding systems using total rRNA sequencing"

_PeerJ, doi:10.7717/peerj.10184_

## Round 0.1 · original submission · Major Revisions

Consider the following comments examples and not a comprehensive list of edits. I give up on edits for style at line 330, as it is inappropriate for me to ghost write this manuscript. and carefully proofread a resubmission. Use the above comments as examples.

Regards,

Michael

Line 38. Here and throughout, write succinctly. For example, replace “The plant biomass breakdown in the rumen depends on the complex microbiota that consists of bacteria, archaea, fungi, and protozoa and their interactions.” With “Breakdown of plant biomass in rumen depends on interactions between bacteria, archaea, fungi, and protozoa.”
Line 40. Again, write succinctly. Replace “… majority of rumen microbiome studies characterized separate microbial groups to understand the microbial fermentation in the gut of the herbivorous animals including camels. This study…” with “the majority of studies of the microbiome of rumininants, including the few studies of the rumen of camels, only studied one of these microbial groups. In this study, we”
Line 42. Here and throughout, write directly and clearly. Replace “get a collective insight into the potential” with “identify”
Line 45. Delete “on”
Line 46. Avoid the passive voice. Replace “.. repectively (spelling). The active bacterial community was the most dominant in the rumen of camel followed by protozoal, archaeal and fungal communities, respectively.” With “Bacteria dominated, followed protozoa, archaea and fungi, libraries of reads generated from the rumen of camels.” See also line 48 (were the most).
Line 47. Avoid fillers like “Our results showed” It should be obvious to the reader when you are presenting your results. See also lines 198, 220 (“in the present study”).
Line 50. Be specific. Replace “groups, where…indices.” with “…groups; libraries generated from camels fed fresh clover showed the highest alpha diversity.”
Line 52-53. Replace “.. showed that the samples of camel groups clustered separately. Variations in the relative abundance of microbial communities across the sample fractions were observed.” With “… showed clusters associated with feeding system and that the relative abundance of microbes varied between liquid and solid fractions of.”
Line 62. Write succinctly. Replace “has high value to support the” with “provides”
Line 64. Repetitive. Replace “In addition to the high….milk. Camels can provide….race riding..” with “Camels also provide textiles…”
Line 72. Avoid superfluous determiner “The” in “The Digestion..” Also, delete “like other ruminants”
Line 74. Delete “in different forms of”
Line 75. Delete “that cooperate…host animal”
Line 77. Delete superfluous “the” and revise to write actively. Replace “shrubs that are mostly avoided by other domestic ruminant” with “shrubs that other domestic ruminants avoid”
Line 83-86. Revise this section.
Line 87. Delete “RNA/DNA”
Line 89. Delete “in the same environment”
Line 92-99. Discuss published work in the present tense. Replace “…outperformed amplicon
Sequencing… chimera structures” with “offers the advantage of specifically targeting active microbes and avoids biases associated with primer selection and chimer generation in PCR”
Line 100. Join this discussion of the benefits of RNAseq with the previous paragraph and replace “it doesn’t just based” with “it doesn’t depend”
Line 145. Here and throughout. Avoid superfluous “The.” Actually delete “The analysis…feeding systems.”
Line 170. Shapiro and Wilk are surnames.
Line 173. Missing a verb.
Line 181-184. Delete this section, which repeats Introduction and Methods.
Line 197-199. Wordy. Replace “… community was similar… and the five most predominant…” with “…community varied little between treatments and consisted of 12 phyla. The five most predominant…”
Line 204. Again, write actively. Replace “this phylum was found to be dominated by six genera” with “six genera dominated this phylum”
Line 227. Revise “were mainly observed in”
Line 242. As above, delete phrases like “The results showed that” and “of the current study” We know results show results. Also, delete “observed in this study (line 247)” and “In this study (line 257).”
Line 267-269. Wordy. Replace “Multivariate analysis …depending on the feeding system.” With “Multivariate analysis separated libraries by feeding system distinctly…”
Line 274-277. Do not repeat Methods. Replace “We performed…metric. The results revealed…” with “PERMANOVA revealed…”
Line 285. As above, revise to “Pearson correlation analysis revealed many…”
Line 303-305. Revise to “Rumen microbes can ferment a wide variety.. and produce volatile fatty acids…”
Line 311. Again with “In this study” and “in the present study”
Line 314. Write actively. Revise to “PCoA…confirmed this finding…”
Line 317. Missing a verb here.
Line 321. Delete “the results obtained from”
Line 326. Replace “have been found to affect negatively the” with “slows”
Line 328. Sample fractions don’t impact the community. Sampling may impact your results.
Line 330. Have a native English speaker read this out loud.
Line 509. Carefully proofread references. (Physiol. Rev.). Italicize genus and species (line 552). Delete “42” line 591. Only cap proper nouns “A proposed taxonomy (line 610).”

Reviewer 1 ·

Basic reporting

Please refer to the attached pdf

Experimental design

Please refer to the attached pdf

Validity of the findings

Please refer to the attached pdf

Additional comments

The manuscript is a welcome addition to the field of the rumen microbiology, describing the functionally active microbial communities in camel. However, the paper needs to be majorly revised.
The introduction and discussion need significant improvement, especially in the use of professional English, connecting various sentences, and discussing obtained results in light of existing knowledge. Some of the sentences end abruptly, without any link to the next sentence, hence, need to be rephrased.
Please refer to the attached pdf for detailed comments.

Annotated reviews are not available for download in order to protect the identity of reviewers who chose to remain anonymous.

Reviewer 2 ·

Basic reporting

The manuscript is fairly well-written, and easy to follow. The background provided is sufficient.

Experimental design

The sample sizes are low, particularly in the wheat straw group, but otherwise the experimental design and analysis of sequences is well described.

Validity of the findings

The findings are justified as presented. The authors state (correctly) that very little work has been done on the camel rumen microbiome, despite their unique physiology and geographic locations, which must affect the rumen microbiota in interesting ways,

Additional comments

line 137: change 'grinded' to 'ground'
line 154: should the eukaryotic RNA be '(18S, 28S)'?

Reviewer 3 ·

Basic reporting

This study reports the analysis of the abundance of rRNA from microbial communities from solid or liquid pseudorumen digesta samples from camels fed on three different diets. The method used here is superior to standard DNA metabarcoding for two reasons: first, analysis of rRNA allows examination of the active microbial community and second, the authors here use RNAseq thus avoiding any primer bias issues associated with metabarcoding approaches were only a small region of the rRNA locus is amplified. In the context of studies of mammalian herbivores this is particularly useful since it allows direct comparison of transcript abundance between the diverse microbial components and also avoids problems of missing species (esp. protozoa) whose rRNA is not amplified well/at all by standard metabarcoding primers.

L48: Use italics for genus/species names (here and later e.g. L62)
L60-70- it is worth reporting in the introduction that one new species of anaerobic fungus was recently discovered from camel and appeared to be specific to camels. Protozoa can show distinct patterns of host specificity, so would be useful to mention if any are distinctive to camles. Useful to link this to the pseudoruminant point below
L72-80: should mention that camels are not ruminant but rather pseudoruminants and discuss potential implications of this difference (ie example of convergent evolution)
L87: Correct to “Most of molecular-based assessments of microbial groups in the rumen have hitherto relied on RNA/DNA-amplicon sequencing”. Here and elsewhere are examples of minor grammatical errors. One final proofreading from native English speaker would help expunge these
L100: as it is not reliant on primers for known microbes
L122: need more details of the feed. For example was G2 all fresh forage; authors state forage diet OR clover. Seems that the camels in this group did not receive a uniform diet. Can they state for each camel what the feed was? and for how long prior to slaughter the camels receive each diet
what was interval between last feeding and slaughter?
L137: were ground using liquid nitrogen

L186: A useful initial figure (clearer than Table 1 would be to show relative abundance of transcripts of the main microbial groups. Fungi are lower than other groups but unexpectedly not more abundant in the solid fraction. Is anything known about relative rRNA abundance per unit active biomass in bacteria vs archaea/eukaryotes. Or do these findings mean that bacteria comprise ca 90% of total microbial biomass in this habitat?
L186-195: Was any rRNA originating from the feed components detected, for example chloroplast rRNA. One treatment contained fresh forage so this would be expected.
L234: Methanobrevibacter
L239: Methanomicrobiales /Methanomicrobiaceae? (need to check the organism names, families etc carefully!)
L241-264: Identification of eukaryotes was done with 18S but for fungi at least this locus shows poor resolution. Use of LSU would provide more accurate taxon identification, and probably for protozoa too.
L261: Odd to see Spizellomyces. Not an anaerobe but is associated with dung sometimes. Maybe was on the ingested forage.
Fig. 2: Some of the labels on the plot are difficult to read- can the plot be re-done to avoid this problem
L45 Fibrobacter spp.

Experimental design

The manuscript describes original primary research within Scope of the journal.
As noted above, the methods are appropriate and innovative. Good that samples were separated into liquid/solid components to allow comparison of planktonic vs attached microbial communities. Similarly bioinformatic approaches are also appropriate.
I assume that experiments were conducted in compliance with ethical /legal guidelines in Egypt.

The main thing that perplexes me with this study us the unusual choice of treatments and the uneven number of replicates (6,3,2; see L123-128). This is odd and is not explained. Having only 2 reps for one treatments greatly constrains subsequent statistical analysis.

I was also unclear why the three treatments were chosen. More precise detail of these is needed as noted above. Why did the authors not just pick simpler diets, e.g. 100% straw, 100% green clover and 100% clover hay. This would have more clearly addressed their first hypothesis (“get insight into the composition of
116 active microbiota in the rumen of camels reared under different feeding systems”)

Validity of the findings

The study is very interesting in that it assesses abundance of active biomass in a more effective way that DNA metabarcoding. It think that the identification of the eukaryote communities would be enhanced by analysis of LSU (28S) transcripts to obtain more accurate taxon identification.

It is also valuable in that the relative proportions of transcripts of members of the bacterial, archaeal, protozoan and fungal communities can been assessed together (and this aspect should be discussed in greater detail in the discussion)

Sadly, the odd experimental design and uneven/low replicate numbers compromises what can be deduced from the feeding treatments but I think the study is nevertheless valuable.

Additional comments

I think that the correct spelling of the 2nd author’s name is “Robert J Forster”

---

## Round 0.2 · Minor Revisions

The paper appears close, in terms of content, but still needs revision for style and format. Please follow these examples.

Line 39. Combine to “protozoa; however, the..”
Line 60. Revise to “Camels (Camelus dromedaries) can produce milk and meat in hot, arid and semi-arid regions and can provide food security as the climate warms…”
Line 62. Revise to “Camels also provide textiles (fiber and hair) and are commonly…
Line 64. Replace “This unique..tract (Kay…” with “The unique feeding behavior and the functional structure of digestive tract of these pseudo-ruminants is well adapted to deserts (Kay..”
Line 66. Revise to “is longer than cows, sheep and other true ruminants, which…”
Line 69. Revise to “feeding type. Camels…”
Line 72. Replace “Many factors a...determiner of the diversity of rumen microbial communities” with “Diet and feeding plan, determine the diversity of rumen microbial communities but age, animal breed can also influence the structure of this microbiome.”
Line 75. Replace “diet is the major shaper of fermentation” with “diet shapes fermentation”
Line 296. Replace “The structure of microbial community in the camel rumen was similar in the composition; however, feeding…” with “The rumen microbiome varied little between animals sampled. As predicted, feeding…”
Line 302. Replace “The Egyptian clover is considered …mixture (Carberry..” with “Egyptian clover is the most balanced and nutritious fodder widely used for feeding camels (Carberry..”
Line 320. Avoid phrases like “are known to” and “found to be” Revise to “genera have a key role as reductive acetogens (Le Van et al., 1998; Yang et al., 2016) and varied with feeding system…”
Line 325. Revise to “…2014b). The phylum was dominated by family Prevotellaceae, which confirms Gharechahi et al. (2015).”
Line 329. Replace “Taken together, we speculate..their molecular roles.” With “We speculate that Bacteroidetes species contribute to the adaptation of camels to arid conditions.”
Line 350. Revise to “…2015) and which..play a major role” with “…2015), which confirms that the attached microbes play a major role..”
Line 352. Leave calls for future work to reviews and proposals. Delete “Further work is needed to examine …the rumen.”
Line 361. Present reasonable significant figures (38% not 38.53%).
Line 362, Revise to “a study of Muskoxen”
Line 366. Again, do not call for more work. Delete “These unclassified bacteria need more studies to enable their isolation and identification.”

Reviewer 1 ·

Basic reporting

Please refer to the attached file.

Experimental design

Please refer to the attached file.

Validity of the findings

Please refer to the attached file.

Annotated reviews are not available for download in order to protect the identity of reviewers who chose to remain anonymous.

---

## Round 0.3 · Minor Revisions

The manuscript appears close but we need to hear from coauthor Forster. It doesn't make sense to invest time in revising a draft of the manuscript that hasn't been approved by all coauthors.
Also, Carefully proof read references for format. For example:

Line 504. “3363±71” ?
Line 525 Fungal Biol. 119:731-737
Line 544 Front Microbiol. 8, e1814
Line 654 Scientific reports, 7, e13047

---

## Round 0.4 · Minor Revisions

I have taken a fresh look at this manuscript, now that the authors have been confirmed. As stated previously, it is not appropriate to ask reviewers to work on a version that the authors couldn’t possibly have read. Going forward, I expect all co-authors to review each version of the manuscript.

On further review, it appears that camels receiving different treatments (feed) were housed in different facilities (line 126). This would make it difficult to differentiate the difference between feed and facility. I do not think this is a fatal case of pseudoreplication (Hurlbert 1984) but I do think Methods have to be clarified and this potential flaw in experimental design, as well as the limited number of samples, needs to be addressed in Discussion.

Figure 2 is not publication quality.
Line 46. Remove extra period.
Line 53. Replace “This study...of the rumen” with something more specific like “This provides preliminary evidence that bacteria dominate the microbial communities of the camel rumen and that feed changes that microbiome.”
Line 62. Delete “daily human activities such as.”
Line 70. Revise to “concentrated”
Line 74. “The chemical..the rumen.”
Line 76. Replace “...sugars are the...favoring the amylolytic” with “sugars favor amylolytic”
Line 77. Present published facts in present tense. Replace “varied” with “varies”
Line 81 -89. This section needs reorganization. Move the last sentence up to the start of the paragraph. Revise to “The efficiency of microbial fermentations in the rumen depends on ...Kamra, 2005). Analysis of these microbial communities could lead to increases in animal productivity and reduction of greenhouse gas emissions ....”
Line 90. Replace “The development of the next-generation sequencing technologies...for the rapid identification...” with “Next generation sequencing technologies provide a rapid method of microbial identification...
Line 108. Replace “All the” with “Previous”
Line 113. Move “Regarding the anaerobic...camel.” to before “Only three molecular-...(line 111).
Line 116. Delete “Moreover, no study provided ....the camel.”
Line 130. Present parallel things in parallel form. “Camels in group G1...” Camels in group G2..” Camels in group G3...”
Line 132-133. These sentences need verbs.
Line 178. Delete “All the”
Line 183. Here and throughout, present reasonable significant figures (359872 ± 85366). See also line 221 (39% not 38.53%) and Table 1, 3 and 4.
Line 224. Replace with “Reads that classified as archaea were further classified to three orders with in the phylum Euryacheota: Thermoplasmatales ..”
Line 228. Replace “All the Methanobacteriale reads were belonged to family” with “Reads that classified in the ... were further classified...”
Line 237. Follow the example provided above for the protozoa (“Reads that classified as protozoa were further...”) and fungi.
Line 310. Replace “was found to be” with “appeared”
Line 311. Delete “with the results of”
Line 329. I stop edits here. The authors must revise this section for style.
Line 470. “Nucleic acids research 37: ?”
Line 491. Why do some references have a period after the page number and some do not?
Line 503. Why do some references provide the issue but most do not?
Line 507, 509, 521, 551.... Cap first letter of journal titles. (Frontiers in microbiology 8:e226, Journal of bacteriology).
Line 531. Is this a journal? Why is it formatted differently and who cares when you accessed it?

---

## Round 0.5 · Minor Revisions

Before I send this to reviewers, I need your group of authors to carefully proofread it. I only briefly reviewed this version and I noticed several minor points, which suggests the manuscript has not been adequately proof read by the coauthors.

My major concern was not addressed. The discussion of the limitations of the experimental design (line 445-449) doesn't make clear that you can tell that the treatment effects were due to feed or location. That should be made clear in lines 293 - 307, with appropriate references on pseudoreplication. You cannot address the flaws in experimental design by simply providing recommendations for the next study.
line 127. Always cap Table and Figures.
Table 2 and throughout. When presenting mean and SE, always present the same significant figures (5 +/- 1 not 5 +/- 0.001).

---

## Round 0.6 · Major Revisions

We received one review recommending acceptance but I still have major concerns about the design of the experiment presented in this manuscript. You cannot distinguish the effect of station and feed. The manuscript must state that clearly and remove all conclusions about the effect of feed. Supplemental note S1 is not appropriate or convincing.

The authors do not appear to have addressed my previous comments about style and format or carefully reviewed the manuscript before submission. The writing needs careful revision using the following examples. This means not just correcting the comment but also looking for similar errors throughout the manuscript. For example, phrases like “Methonomicrobium has been shown to be responsible for the conversion of.. (line 379)” are wordy. The reader can presume that scientific facts come from previous scientific publications, as indicated by the citation. Replace with “Methonomicrobium converts..” and remove phrases like “previous studies showed that (line 393)” and “the study of (line 394)” at those lines and throughout the manuscript.

Line 42. Revise to “These camels were reared at three stations that use different feeding systems: clover, hay and wheat straw (G1), fresh clover (G2), and wheat straw (G3).
Line 48. Delete “Feeding system…microbial groups.” You cannot say if it was feed or facility. Here and throughout, revise to statements like “..camels reared at facility G2, where they were fed fresh clover, showed…”
Line 51. Revise to “…with facility/feed and…”
Line 52. The statement “..analysis showed positive and negative correlations…” adds little. Provide an interesting example or delete.
Line 54. Delete conclusion about feed type.
Line 63. Here and throughout, never use “which” twice in a sentence. Revise to “…microorganisms. This long retention improves the efficiency of digestion..”
Line 65. This paragraph needs reorganization. Delete “Based on …. Concentrated supplements.” Start paragraph with “The feed ranchers provide camels, which ranges from forage in traditional pastures to concentrated supplements in intensive feedlots, influences the structure of the camel microbiome (Henderson et al. 2015)."
Line 76. Start a new paragraph with “Digestion in the camel…
Line 85. Delete “of PCR-based”
Line 89-92. Move “The recent development…culture-based methods..” to start of paragraph.
Line 111. Delete “reared under different feeding systems”
Line 119. Replace “under three different feeding systems” with “reared at three stations that use different feeding systems”
Line 122. Provide the location of the farm, as you did for G1. I suggest “Camels in group G2 were housed at the xxxx Station and fed fresh..
Line 124. As above, revise to “…G3 were housed xxxx and fed wheat…”
Line 202. Delete “The other phyla including” and replace “that was” with “were”
Line 296. Revise to “Camels groups were fed…” Also, as stated above, the experimental design does not allow you to conclude that “diet type has the main effect.” Delete “However…In addition (line 296-298).”
Line 301. Replace “confirmed the finding of this study and”
Line 307. Avoid superfluous determiners. Delete “the” here, in line 309 and 310. Search for this article (the) throughout text and determine if it is necessary.
Line 313. Delete “was”
Line 319. Delete “that was found to be”
Line 324. Revise to “..higher in samples collected from animals reared in the station that used low-quality feed..”
Line 333. Replace “are specialized” with “specialize”
Line 356. Here and throughout, use reasonable and consistent significant figures (54% not 53.7%)
Line 363. Replace “is used to provide” with “provides”
Line 364. Avoid the passive voice. Revise to “alteration of diet shifts the structure of methanogen populations…”
Line 370. Revise to “produce”
Line 379. Replace “has been shown to be responsible for the conversion of” with “converts”
Line 381. Always add a comma before “which”
Line 393. Replace “previous studies …dominant in” with “This genus predominates rumen”
Line 394. Delete “the study of”
Line 416. Avoid the passive voice. Revise to “contamination of forage… could explain the presence…”
Line 419. Delete “the” and replace “are the main driver” with “drive”
Line 423. Delete “was believed to,” unless you no longer believe it.
Line 438. Delete “known to be”
Line 448. Conclusions should provide clear statements of what you observed not what you didn’t do. Delete “This study applied…of our study (line 453).”
Line 454. Remove conclusions about the effect of feed.
Line 467, 470, 473,....As stated in my previous comments, cap the first letter of journal titles. (Journal of Crop Science not Journal of crop science). This comment applies to all references.

Table 1. Be consistent in significant figures, as below.
Archaea SF 2.3±0.2, 3.4±0.4, 2.2±1.0, 3.0±0.3
Archaea LF 2.2±0.2, 2.8±0.4, 1.8± 0.2, 2.0± 0.3

Table 2, 3, 4 and 5. Again, present significant figures. 63±2…

---

## Author Rebuttal · Round 0.6

**Ministry of Agriculture and Land Reclamation**

**Desert Research Center**

[Figure]

وزارة الزراعة واستصلاح الأراضى

مركز بحوث الصحراء

Desert Research Center                                            23ᵗʰ June, 2020

1Mathaf El Matariya St.B.O.P.11753

Matariya- Cairo,Egypt

Phone: (+202)26332846 - 26374800

FAX: (+202) 26357858

Email:Rabee_a_m@yahoo.com

## Dear Editor,

We thank you for your comments. We responded to your comments and enclosed unclean paper including all comments colored by yellow. Below are the responses to all the comments.

We appreciate the opportunity to submit our manuscript to Peer J.

Yours sincerely,

Dr. Alaa Rabee

Researcher at Desert Research Center, Egypt

On behalf of all authors

# Comments

Manuscript title: **"Comparative analysis of the metabolically active microbial communities in the rumen of dromedary camels under different feeding systems using total rRNA sequencing "**

## Editor's comments

**Before I send this to reviewers, I need your group of authors to carefully proofread it. I only briefly reviewed this version and I noticed several minor points, which suggests the manuscript has not been adequately proof read by the coauthors.**

>> We have revised it.

**My major concern was not addressed. The discussion of the limitations of the experimental design (line 445-449) doesn't make clear that you can tell that the treatment effects were due to feed or location. That should be made clear in lines 293 - 307, with appropriate references on pseudoreplication. You cannot address the flaws in experimental design by simply providing recommendations for the next study.**

>> We modified this paragraph at the beginning of the discussion and removed the paragraph at the end of the discussion and make minor change in the conclusion. Furthermore, we included a supplementary not S1, which clarify why the diet has the major impact on the microbial communities, thank you.

**line 127. Always cap Table and Figures.**

>> Modified

**Table 2 and throughout. When presenting mean and SE, always present the same significant figures (5 +/- 1 not 5 +/- 0.001).**

>> Modified.

---

## Round 0.7 · Minor Revisions

The manuscript is improved, particularly in terms of drawing conclusions about the effect of feed. I have minor revisions for style.
Line 42. Delete “reared under three feeding systems”
Line 44. Add space before “Bacteria” and “Camels (line 287)”
Line 116. Revise to “…animals were kept on these diets..”
Line 200. The statement “All Bacterial genera were observed,” suggests you observed all known bacterial genera, which doesn’t seem likely. Provide the total number of genera and revise to “Of the ___ genera observer, only seven were observed exclusively in libraries generated from a specific facility.”
Line 278. Delete “the”.
Line 281. Replace “Therefore, investigation of these microbial communities is the key to understand their roles and maximize ruminal fermentation and fiber digestion and reduction of greenhouse gas” with “Investigation of these microbial communities could improve our understanding of their function in fiber digestion and lead to practices that maximize the efficiency of ruminal fermentation and minimize greenhouse gas release.”
Line 299. Replace “is in agreement with” with “agrees with”
Line 303. Avoid the passive voice. Replace “This speculation was supported by the high proportion of Ruminococcaceae and Lachnospiraceae…” with “The high proportion of Ruminococcaceae and Lachnospiraceae supports this speculation.”
Line 304. Delete “Bothe”
Line 307. Replace “This finding could indicate that the reductive acetogenesis pathway could be maximized by diet to minimize methane...” With “This suggests that manipulation of diet can enhance reductive acetogenesis in rumen and minimize methanogenesis…”
Line 345. Replace “The archaeal population has important roles in methane emission mitigation strategies as they convert the H2 and CO2 produced in the rumen to methane.” With “Since some archaea produce CH4 from H2 and CO2, this phyla may control methane emission from ruminants.”
Line 367. Replace “were related to” with “were classified as”
Line 369. Here and throughout, write actively. Replace “The relative abundance of protozoal was influenced by feeding system and housing facility, which was in the same line with results on cattle” with “Feed appeared to influence the relative abundance of protozoa, as reported previously for cattle (Hristov et al., 2001; Weimer, 2015); however, we cannot differentiate the effects of feed from facility.”
Line 371. Here and throughout, avoid superfluous determiners. Delete “The”
Line 374. Here and throughout, delete phrases like “is considered to be,” unless you doubt that fact. Replace with such as Diplodinium cameli (Kubesy and Dehority, 2002) “some species of genus Diplodinium, such as Diplodinium cameli, were discovered in, and are unique to, the rumen of Egyptian camel.”
Line 388. Delete “This genus”
Line 508. Delete “Available at: http://livestocklibrary.com.au/handle/1234/20056. Accessed June 8, 2020”
Line 533. Volume is 2010.
Line 667. Page number is e000066
Table 1. Be consistent with significant figures. 89±2 not 88.5±2.
Table 2. Be consistent with significant figures. 0.6±0.3 not 0.6±0.25

---

## Round 0.8 · Minor Revisions

The manuscript still requires revisions for style. These changes follow recommendations from previous reviews I provided; however, in those previous reviews I asked that the authors followed these examples throughout, which hasn’t happened. I add some more examples below and ask that all authors carefully follow these suggestions throughout.
Do not just do the following changes. Use these examples as templates.

Line 38. Avoid superfluous determiners. Delete “The” in “The breakdown…” and replace “the camel” with “camels” (line 68).
Line 51. On further review, something is missing here. I suggest revising to “...bacteria dominate the microbial communities of the camel rumen and these communities varies between populations of domesticated camels.”
Line 82. Avoid the passive voice. Replace “Therefore, the output could be biased due to the primer selection and amplification cycling conditions” with “Primer selection and amplification conditions could bias the output…” See also lines 182, 187, 189, 247, 350…
Line 104. Something is missing here. I suggest revising to “… rumen fractions; 3) assessing the heterogeneity of these microbial populations within different populations of domestic camels.”
Line 180. Delete “Phylum”
Line 182. Again, here and throughout, avoid the passive voice. Replace “On the family level, the Firmicutes phylum was dominated by Lachnospiraceae and Ruminococcuceae.” With “At the family level, Lachnospiraceae and Ruminococcuceae dominated the Firmicutes.”
Line 247. Replace “..bacteria, dominated by phylum Firmicutes were the main driver of differences…” with “bacteria, dominated by Firmicutes, drove differences…” See also line 248.
Line 282. Revise to “In this study, camel groups..”
Line 284. Revise to “…with the results of studies of other ruminants..”
Line 298. Why cap “Muskoxen” (see also line 337)?
Line 304. Revise to “This supports the observation that manipulation…”
Line 321. Delete “as is was reported by”
Line 330. Replace “Most of Elusimicrobia in this study were uncultured” with “Most of Elusimicrobia observed in this study have yet to be cultured…”
Line 346. Revise to “that were fed”
Line 350. Replace “and its population was decreased by the addition of rapeseed oil to animal diet” with “and supplementing of animal’s diet with rapeseed oil decreases the abundance of this order.”
Line 352. Delete “The”
Line 353. Present your results in past tense. Revise to “…(Table 4) dominated the reads classified as Archaea in this study, which agrees with trends reported for beef cattle..”
Line 358. Why cap “In vitro”?
Line 382. Delete “speculation”
Line 391. Replace “Therefore, the fungi were more prevalent in ruminants of G2 camels, which were fed high-quality forage with high fiber contents than in G1 and G3 camels” with “Fungi were most abundant in rumen collected from the G2 group of camels.”
Line 395. Delete “the” in “the forages” and “the camel”.
Line 400, 407. Delete “the”
Line 401. Delete “which highlighted some positive correlations between protozoa and archaea.”
Line 402. Delete “important substrates mainly”
Line 414. You cannot have to phrases that start with “which” in the same sentence.
Line 416. Replace “fungi are negatively impacted by the presence of some bacteria and protozoa as the fungal zoospores are likely to be a prey for protozoa” with “some bacteria and protozoa prey on fungal zoospores”
Line 420. Revise to “which supports”
Line 423. Again, too many dangling phrases. Delete “which might…Protozoa.”
Line 425. Revise “between groups of camels.”
Line 429. Replace “camel found to be similar to other ruminants with a shown difference in the relative abundances” with “camels was similar to other ruminants.”
Line 430. Delete “The present…their functions.”

---

## Round 0.9 · accepted · Accept

The manuscript is acceptable with minor revisions for style. I leave that to the authors and production editors.
Line 52. “variy” is not spelled correctly. Replaced with “differ significantly”
Line 55. Delete “can” in “can provide”
Line 57. Replace “race and riding” with “racing”
Line 59. Since camels are pseudoruminants, the phrase “other true ruminants” doesn’t make sense. Change to “than the retention time for true ruminants...” Also, delete “prolongs the exposure of plant biomasses to the symbiotic microorganisms. This long retention...” so that it reads “... which improves the efficiency...”
Line 279. Replace “Furthermore, rumen fermentation generates methane” with “Fermentation by rumen microbes also generates methane...”
Line 300. Replace “on” with “of”